# Hypothermia Advocates Functional Mitochondria and Alleviates Oxidative Stress to Combat Acetaminophen-Induced Hepatotoxicity

**DOI:** 10.3390/cells9112354

**Published:** 2020-10-26

**Authors:** Yeong Lan Tan, Han Kiat Ho

**Affiliations:** 1Department of Pharmacy, Faculty of Science, National University of Singapore, Singapore 117543, Singapore; yeonglan_tan@u.nus.edu; 2NUS Graduate School for Integrative Sciences & Engineering, Centre for Life Sciences, National University of Singapore, Singapore 119077, Singapore

**Keywords:** drug-induced liver injury, hepatotoxicity, hypothermia, cooling, acetaminophen

## Abstract

For years, moderate hypothermia (32 °C) has been proposed as an unorthodox therapy for liver injuries, with proven hepatoprotective potential. Yet, limited mechanistic understanding has largely denied its acceptance over conventional pharmaceuticals for hepatoprotection. Today, facing a high prevalence of acetaminophen-induced liver injury (AILI) which accounts for the highest incidence of acute liver failure, hypothermia was evaluated as a potential therapy to combat AILI. For which, transforming growth factor-α transgenic mouse hepatocytes (TAMH) were subjected to concomitant 5 mM acetaminophen toxicity and moderate hypothermic conditioning for 24 h. Thereafter, its impact on mitophagy, mitochondrial biogenesis, glutathione homeostasis and c-Jun N-terminal kinase (JNK) signaling pathways were investigated. In the presence of AILI, hypothermia displayed simultaneous mitophagy and mitochondrial biogenesis to conserve functional mitochondria. Furthermore, antioxidant response was apparent with higher glutathione recycling and repressed JNK activation. These effects were, however, unremarkable with hypothermia alone without liver injury. This may suggest an adaptive response of hypothermia only to the injured sites, rendering it favorable as a potential targeted therapy. In fact, its cytoprotective effects were displayed in other DILI of similar pathology as acetaminophen i.e., valproate- and diclofenac-induced liver injury and this further corroborates the mechanistic findings of hypothermic actions on AILI.

## 1. Introduction

The discovery of acetaminophen (APAP), as an analgesic and antipyretic agent, can be dated back to the 1960s [1]. For more than five decades, despite an agglomeration of medications emerging with similar indications, APAP has remained as the most commonly used medication in the United States [2]. Yet, its rampant and unrestricted use has, inevitably, raised a safety concern on its potential for hepatotoxicity. Hitherto, APAP-induced liver injury (AILI), among other pharmaceuticals, has engendered the highest incidence of acute liver failure (ALF) in Western countries [3]. With an alarming mortality rate of close to 50% [4], coupled with limited treatment options [5], the distressing notion of ALF has driven countless research on AILI over the years to uncover mechanistic insights and discover novel effective management strategies.

Today, the broad understanding on the toxico-pathological mechanism of AILI involves two key aspects—mitochondrial dysfunction and oxidative stress. These cellular perturbations are initiated by an accumulation of the reactive metabolite, N-acetyl-p-benzoquinone imine (NAPQI), following the metabolism of APAP in the liver [6]. While NAPQI can be detoxified with reduced glutathione (GSH), excessive NAPQI can deplete the cellular reservoir of GSH and renders the downstream toxic effects, central in the mitochondria [7]. Specifically, NAPQI can interact with proteins and form mitochondrial protein adducts leading to mitochondrial damage; while oxidative stress can also arise from these effects directly or indirectly via GSH depletion [7,8]. Furthermore, the activation and mitochondrial translocation of c-Jun N-terminal kinase (JNK) has also been implicated in the subsequent amplification of oxidative stress [9,10,11]. All these, collectively, result in extensive necrotic cell death.

Currently, the only approved treatment for AILI is via the administration of N-acetyl cysteine (NAC) [6]. Being synthetic precursors of GSH, NAC promotes GSH repletion for detoxification of excess NAPQI and scavenging of reactive oxygen species (ROS) [7,12]. Other proposed pharmacological options also include the potential use of antioxidants or JNK inhibitors to mitigate oxidative stress-mediated damage [9,13]. To limit mitochondrial dysfunction, the removal or replacement of damaged mitochondria through mitophagy or mitochondrial biogenesis could plausibly allay APAP-induced hepatotoxicity as well [10]. Interestingly, the use of hypothermic conditioning, as a non-pharmacological approach, has also been proposed to be a viable option. By marginally downregulating physiological body temperature (36.5–37.5 °C) to as low as 32 °C, Vaquero et al. has demonstrated the possibility of employing moderate hypothermia for attenuating AILI in mice [14]. Its efficacy was, similarly, observed in other liver injuries too, with a remarkable reduction in serum alanine aminotransferases and lesser hepatic damage [15,16]. Even in the late stage liver injury, Zhu et al. has also displayed desirable outcomes of moderate hypothermia in ALF in vivo [17].

Notwithstanding these intriguing findings, hypothermic conditioning today remains an unorthodox therapy, in contrast to targeted pharmaceutics. On hindsight, the reservations may stem from its unexplored mechanism in liver injury. Unlike ischemic injuries where hypothermia exhibits proven antioxidant and promitochondrial properties [18,19,20,21], it remains to be determined if similar actions of hypothermia would be observed in AILI. Henceforth, we conducted an in vitro mechanistic investigation to elucidate the protective effects of hypothermia in APAP-induced hepatocellular injury (AIHI). Specifically, we evaluated the capacity of moderate hypothermia to ameliorate mitochondrial dysfunction and oxidative damage, which are pertinent to elicit AIHI. To do so, we adhered to the optimized hypothermic condition unraveled in our past work—based on changes in the cell viability and cell death profile, moderate hypothermia (32 °C) conditioned for 24 h concomitant with 5 mM APAP toxicity ensued optimal cytoprotection in AIHI [22]. By expounding on the mechanistic effects of hypothermia, we hope to enhance its therapeutic value in AILI, and furthermore, in other liver injuries of similar pathologies. Clinically, we envisage the employment of hypothermic conditioning, during the onset of AILI, to alleviate the detrimental effects associated with APAP intoxication.

## 2. Materials and Methods

### 2.1. Cell Culture and Reagents

Transforming growth factor-α transgenic mouse hepatocytes, TAMH, (a kind gift from Professor Nelson Fausto, University of Washington) were cultured in Dulbecco’s modified eagle medium: nutrient mixture F-12 (Life Technologies, Carlsbad, CA, USA) supplemented with 10 mM nicotinamide (Sigma-Aldrich, St. Louis, MO, USA), 100 nM dexamethasone (Sigma-Aldrich, St. Louis, MO, USA), 2 µg/mL of gentamicin (Sigma-Aldrich, St. Louis, MO, USA) and 0.1% (*v*/*v*) of insulin-selenium-transferrin supplement (Bio Laboratories, Singapore, Singapore) at 37 °C in a humidified atmosphere containing 5% CO_2_, based on Wu et al. [23]. Human liver hepatocytes, L-02, (a kind gift from Associate Professor Yu Chun Kong Victor, National University of Singapore) were cultured in Dulbecco’s modified eagle medium (Sigma-Aldrich, St. Louis, MO, USA) supplemented with 10% (*v*/*v*) fetal bovine serum (GE Healthcare, Pittsburgh, PA, USA) at 37 °C in a humidified atmosphere containing 5% CO_2_. APAP (Sigma-Aldrich, St. Louis, MO, USA) was prepared in PBS at 40 mM and diluted with fresh media into 5 mM; diclofenac sodium (Sigma-Aldrich, St. Louis, MO, USA) was prepared in dimethyl sulfoxide at 100 mM and diluted with fresh media into 2.5 mM while sodium valproate (Tokyo Chemical Industry, Tokyo, Japan) was prepared in ultrapure water at 100 mM and diluted with serum-free media into 10 mM. Anti-AMPKα, anti-phospho-AMPKα (Thr172), anti-p70S6K, anti-phospho-p70S6K (Thr389), anti-ULK1, anti-phospho-ULK1 (Ser757), anti-phospho-ULK1 (Ser317), anti-Beclin-1, anti-phospho-Beclin-1(Ser93), anti-JNK, anti-phospho-JNK (Thr183/Tyr185) and anti-GAPDH antibodies were purchased from Cell Signaling (Danvers, MA, USA); anti-LC3B, anti-VDAC1, anti-3-nitrotyrosine and anti-PGC-1α antibodies were purchased from Abcam (Cambridge, UK), while anti-β-actin, anti-TOM20 and anti-TFAM were obtained from Santa Cruz Biotechnology (Dallas, TX, USA). The horseradish peroxidase (HRP)-conjugated antirabbit and antimouse secondary antibodies were purchased from Cell Signaling (Danvers, MA, USA). All other reagents were obtained from Sigma-Aldrich (St. Louis, MO, USA) or Thermo Scientific (Waltham, MA, USA).

### 2.2. Hypothermic Conditioning

TAMH or L-02 was conditioned with moderate hypothermia by incubating cells at 32 °C for 24 h in a humidified atmosphere containing 5% CO_2_. This was performed concomitantly while cells were subjected to 24 h of 5 mM APAP toxicity in TAMH, or 2.5 mM diclofenac or 10 mM sodium valproate toxicity in L-02.

### 2.3. Immunofluorescence Imaging

1 × 10^5^ cells/well were seeded on coverslips in 12-well plates for 24 h. To demonstrate mitophagy, prior to APAP and hypothermic treatment, TAMH was pretreated with an autophagy inhibitor, 80 µM chloroquine (CQ), for 2 h. Thereafter, cells were treated with APAP at 32 °C for 24 h, together with a concomitant CQ treatment for 24 h. On the other hand, to demonstrate the concomitant occurrence of mitophagy and mitochondrial biogenesis, TAMH was treated with 250 nM MitoTracker^®^ Red CM-H2XRos (Invitrogen, Carlsbad, CA, USA) for 30 min at 37 °C prior to APAP and hypothermic treatment for 24 h. Following on, cells were fixed with 4% (*w*/*v*) paraformaldehyde (Sigma-Aldrich, St. Louis, MO, USA) for 15 min and permeabilized with 0.1% (*v*/*v*) Triton X-100 (Sigma-Aldrich, St. Louis, MO, USA) for 3 min at room temperature. Next, blocking of non-specific binding sites was performed with 2% (*w*/*v*) bovine serum albumin (BSA) in PBS for an hour at room temperature before incubating with primary antibodies diluted in 1:200 with 2% BSA (*w*/*v*) in PBS for another hour at room temperature. Cells were then washed thrice with PBS and incubated with Alexa Fluor 488 and/or Alexa Fluor 594 (Thermo Scientific, Waltham, MA, USA) secondary antibodies for an hour at room temperature. This was followed by 3 PBS washes, where cells were incubated with Hoechst 33342 dye (Sigma-Aldrich, St. Louis, MO, USA) during the second wash. Finally, coverslips were mounted with mounting medium (Sigma-Aldrich, St. Louis, MO, USA) and viewed with an Olympus Fluoview FV10i confocal microscope (Olympus, Tokyo, Japan).

### 2.4. Measurement of Mitochondrial Membrane Potential

2 × 10^5^ cells/well were seeded on 6-well plates for 24 h prior to APAP treatment at 32 °C for 24 h. Thereafter, cells were incubated with 250 nM tetramethylrhodamine methyl ester (TMRM) (Thermo Scientific, Waltham, MA, USA) at 37 °C for 30 min. Following on, cells were trypsinized and resuspended in PBS. Mitochondrial membrane potential (ΔΨM) was measured by flow cytometry using cytoFLEX (Beckman Coulter, Brea, CA, USA) based on 20,000 events. Cells treated with 10 µM carbonyl cyanide-p-trifluoromethoxyphenylhydrazone (FCCP) (Sigma-Aldrich, St. Louis, MO, USA) for 2 h at 37 °C were used as positive control and the relative levels of ΔΨM, in percentage, were determined by normalizing against the negative control i.e., APAP-free treatment with continuous 37 °C incubation. A total of 3 biological replicates were performed.

### 2.5. Isolation of Mitochondrial and Cytosolic Fraction

Cells were seeded on 10 cm dish to reach 90% confluence prior to APAP treatment at 32 °C for 24 h and the isolation of mitochondrial and cytosolic fraction was performed as previously described [24]. Briefly, cells were homogenized in mitochondrial isolation buffer (pH 7.5) comprising 50 mM HEPES, 320 mM sucrose, 10 mM potassium chloride, 1.5 mM magnesium chloride, 1 mM ethylenediaminetetraacetic acid, 1 mM dithiothreitol and protease inhibitor cocktail which included 10 mM sodium fluoride, 100 mM phenylmethylsulfonyl fluoride, 2 mM sodium orthovanadate and 2 μg/mL aprotinin. Homogenization was performed mechanically with a 25 G needle using a 1 mL syringe. Thereafter, cell debris was removed by centrifuging cell lysates at 800× *g* for 10 min. The supernatant was then collected and further centrifuged at 10,000× *g* for 20 min. Here, the supernatant was collected as the cytosolic fraction while the resultant pellet forms the mitochondrial fraction. Finally, the pellet was resuspended in cell lysis buffer comprising of the abovementioned protease inhibitor cocktail, together with 1% (*v*/*v*) octylphenoxypolyethoxyethanol, 0.5% (*w*/*v*) sodium deoxycholate and 0.1% (*w*/*v*) sodium dodecyl sulfate (SDS) diluted in PBS. All chemicals stated were obtained from Sigma-Aldrich (St. Louis, MO, USA).

### 2.6. Western Blotting

2 × 10^5^ cells/well were seeded on 6-well plates for 24 h prior to APAP treatment at 32 °C for 24 h. Following hypothermic and APAP treatment, cell samples were lysed in cell lysis buffer stated in the previous section. 20 μg of protein samples were separated via 8–15% (*v*/*v*) SDS-polyacrylamide gel electrophoresis and transferred onto polyvinylidene fluoride membranes (Bio-Rad, Hercules, CA, USA). Following on, all membranes were blocked with 5% (*w*/*v*) BSA in tris-buffered saline containing 0.1% (*v*/*v*) Tween-20 (TBS-T) for an hour with the exception of assay with anti-3-nitrotyrosine antibodies, which was blocked with 5% skim milk in TBS-T for 3 h. Thereafter, all membranes were incubated overnight at 4 °C with primary antibodies diluted in 1:1000, except anti-β-actin and anti-GAPDH antibody which were diluted in 1:10,000. Next, all membranes were washed with TBS-T before incubating with HRP-conjugated secondary antibodies, at 1:10,000 dilution, for an hour. The protein bands were then visualized with chemiluminescence image analyzer (G:BOX Chemi XX6, Syngene, Cambridge, UK) using western lightning plus-ECL reagent (PerkinElmer, Waltham, MA, USA). β-actin was used as the housekeeping protein for most blots while GAPDH and VDAC1 were used as the housekeeping protein for cytosolic and mitochondrial fractions respectively in samples involving subcellular fractionation.

### 2.7. Real-Time PCR Analysis

2 × 10^5^ cells/well were seeded on 6-well plates for 24 h prior to APAP treatment at 32 °C for 24 h. Following hypothermia and APAP treatment, total RNA was extracted using RNeasy mini kit (Qiagen, Venlo, NL, USA) while total DNA was extracted using DNeasy blood and tissue kit (Qiagen, Venlo, NL). Thereafter, RNA and DNA content were quantified with NanoDrop 1000 UV/Vis spectrophotometer (Thermo Scientific, Waltham, MA, USA) and cDNA was synthesized from 1 µg of total RNA using qScript cDNA SuperMix (Quantabio, Beverly, MAk, USA) based on manufacturer’s recommendations. The list of primers (Integrated DNA Technologies Coralville, IA, USA) used are shown in Table 1 and β-actin or hexokinase 2 was used as the housekeeping gene. Real-time polymerase chain reaction (PCR) analysis was performed with QuantiFast SYBR Green PCR Kit (Qiagen, Beverly, MA, USA) on CFX96 touch real-time PCR detection system (Bio-Rad, Hercules, CA, USA) and the cycling conditions were as follow–samples were heated at 95 °C for 5 min, followed by 40 cycles of 95 °C for 10 s and 60 °C for 30 s. The relative mRNA expressions or DNA content were determined based on fold changes calculated using 2-ΔΔCt, where normalization was performed against the negative control i.e., APAP-free treatment with continuous 37 °C incubation. All samples were run in triplicates and 3 biological replicates were performed.

### 2.8. Measurement of Hepatic GSH Content and GSH/GSSG Ratio

1.5 × 10^4^ cells/well were seeded on 96-well plates for 24 h prior to APAP treatment at 32 °C for 24 h. Thereafter, to determine the relative hepatic GSH content in TAMH, both live and dead cells were harvested and glutathione cell-based detection kit (Cayman Chemical, Michigan, USA) was used according to manufacturer’s instructions. By using monochlorobimane as the GSH substrate, it reacted with GSH and the resultant fluorescence intensity was measured at 380 nm excitation/480 nm emission wavelength on a microplate reader (Hidex, Turku, Finland). Relative changes, by percentage, in the GSH content were determined by normalizing against the negative control i.e., APAP-free treatment with continuous 37 °C incubation. A total of 3 biological replicates was performed.

To further examine the extent of reduced (GSH) and oxidized glutathione (GSSG) levels, 1.5 × 10^4^ cells/well were seeded on 96-well plates for 24 h prior to APAP treatment at 32 °C for 24 h. Next, both the live and dead cells were harvested and the GSH/GSSG ratio detection assay kit (Abcam, Cambridge, UK) was used according to manufacturer’s instructions. With the use of thiol green fluorophore, the fluorescence intensity representing the total glutathione and reduced GSH were measured at 490 nm excitation/520 nm emission wavelength on a microplate reader. Thereafter, the level of GSSG was calculated and GSH/GSSG ratio was determined. All samples were run in duplicates and 3 biological replicates were performed.

### 2.9. Cell Viability Assay

1 × 10^4^ cells/well were seeded on 96-well white plates for 24 h prior to diclofenac or sodium valproate treatment at 32 °C for 24 h. Thereafter, CellTiter-Glo luminescent cell viability assay (Promega, Madison, WI, USA) was used according to manufacturer’s instructions. Briefly, CellTiter-Glo reagent was added in the same volume as the culture medium in each well after hypothermic and drug treatment. Following on, the plates were placed on an orbital shaker at room temperature for 10 min before measuring luminescence intensity with a microplate reader. The relative percentage changes in luminescence reading were determined by normalizing against the negative control i.e., drug-free treatment with continuous 37 °C incubation. All samples were run in triplicates and 3 biological replicates were performed.

### 2.10. Cell Death Analysis

2 × 10^5^ cells/well were seeded on 6-well plates for 24 h prior to diclofenac or sodium valproate treatment at 32 °C for 24 h. Following on, both live and dead cells were harvested and stained with 10 μg/mL propidium iodide (Sigma-Aldrich, St. Louis, MO, USA), diluted in PBS. The percentage of dead cells were determined with flow cytometry based on 20,000 events and a total of 3 biological replicates was performed.

### 2.11. Statistical Analysis

All data were expressed as mean ± SD of 3 biological replicates. Statistical analysis was conducted using GraphPad Prism for Windows (version 7.00) (GraphPad Software, La Jolla, CA, USA). Unpaired Student’s *t*-test was performed for all statistical analysis and differences between groups were considered statistically significant for *p* < 0.05.

## 3. Results

### 3.1. Moderate Hypothermic (32 °C) Conditioning Promotes ULK1-Independent Mitophagy via AMPKα Activation in the Presence of AIHI

To combat AIHI, the removal of damaged mitochondria by autophagy has been frequently reported [26,27]. Since hypothermic conditioning was shown to promote autophagy in other studies [28], we, therefore, began our investigation to pursue the plausible role of autophagy in hypothermic protection against AIHI in TAMH. Indeed, with moderate hypothermia, an increase in microtubule-associated proteins 1A/1B light chain 3B (LC3B)-II/β-actin ratio was observed in the presence of CQ, a lysosomal inhibitor, denoting the presence of autophagy (Figure 1A). With AIHI, the heightened autophagy was accompanied with AMP-activated protein kinase (AMPK) activation alongside downstream activation of Beclin-1 complex in the autophagy signaling pathway (Figure 1D). Together with a concomitant suppression of unc-51-like autophagy activating kinase 1 (ULK1) (Ser757) phosphorylation (Figure 1D), the AMPK-mediated autophagy has seemingly occurred in a ULK1-independent manner, notwithstanding a repressed transcriptional state of AMPK following hypothermia in AIHI (Figure 1B). In contrast, mechanistic target of rapamycin complex 1 (mTORC1), as a negative regulator of autophagy, does not advocate autophagic flux during hypothermia despite its suppressed transcript expressions and activity (Figure 1C,D). The unchanged level of ULK1 (Ser317) phosphorylation further corroborates a ULK1-independent autophagy, without mTORC1 regulation (Figure 1D). Interestingly, all these signaling pathways are sensitized by hypothermia only in the presence of concomitant APAP toxicity. With hypothermia alone, an autophagic flux could occur independently from AMPK and mTORC1 regulation (Figure 1B–D).

In relation to AIHI, the phenomenon of autophagic flux was further explored in its role in removing damaged mitochondria, otherwise known as mitophagy. To do so, one of the mitochondrial proteins within the translocases of outer membrane (TOMs) complex i.e., TOM20 was used as a surrogate marker for mitochondria. With immunofluorescence imaging, the apparent overlay between TOM20 and the microtubules associated with the autophagosome membrane i.e., LC3B, illustrated an internalization of mitochondria within the autophagosome prior to its degradation, following hypothermia (Figure 1E). Similarly, the accumulation of TOM20 proteins ensuing an autophagic inhibition with CQ, further supports the notion of mitophagy induced with hypothermia (Figure 1A). In fact, this has been observed to occur with moderate hypothermic conditioning, regardless of APAP toxicity.

### 3.2. Moderate Hypothermic (32 °C) Conditioning Fosters Mitochondrial Biogenesis in the Presence of AIHI

Beyond mitophagy, the investigation proceeded to explore mitochondrial biogenesis in TAMH in order to provide a more holistic understanding of its overall health following hypothermia in AIHI. Accordingly, an increased protein expression of peroxisome proliferator-activated receptor gamma coactivator 1-alpha (PGC-1α), the master regulator of mitochondrial biogenesis [29,30], was observed and hypothermia appeared to foster the proliferation of mitochondria (Figure 2A). Yet, exclusive to the presence of APAP toxicity, the downstream effects of PGC-1α, a coactivator of nuclear receptors, became distinct. There was an increased transcription of transcription factor A mitochondrial (TFAM) followed by its translocation into mitochondria to promote mitochondrial DNA replication (Figure 2B,C). Collectively, these data demonstrated de novo mitochondrial biogenesis to boost the mitochondrial DNA levels, rather than a passive accumulation of mitochondrial DNA fragments, such as in the case of cells subjected to APAP intoxication [31]. Therefore, hypothermic conditioning may augment mitochondrial biogenesis in an adaptive manner following AIHI. Of note, this has attenuated the overall mitochondrial damage, with lesser nitrotyrosine protein adducts in AIHI, albeit an unremarkable change in mitochondrial activity (Figure 2D,E).

### 3.3. Moderate Hypothermic (32 °C) Conditioning Renders a Concomitant Interplay of Mitophagy and Mitochondrial Biogenesis During AIHI

Following an independent demonstration of mitophagy and mitochondrial biogenesis in AIHI upon hypothermic conditioning, we would, next, like to affirm their concomitant occurrence in mitigating hepatotoxicity. To do so, we employed MitoTracker^®^ Red CM-H2XRos to track the live mitochondria preceding APAP and/or hypothermic treatment in TAMH. Thereafter, TOM20 was used as the surrogate marker of mitochondria to track the total mitochondria levels at the end of APAP toxicity for 24 h. The overlay of these fluorescent signals would, therefore, represent the mitochondria of cells which survived APAP toxicity while the fluorescent signal from TOM20 alone, without overlay, would suggest new mitochondria biosynthesis. As such, the changes in these signals would provide an indication of the dynamic changes in mitochondrial count following APAP and/or hypothermic treatment. Based on the immunofluorescence imaging, there was a reduced intensity of MitoTracker^®^ Red and a reduced overlay between MitoTracker^®^ Red and TOM20 after hypothermic conditioning in AIHI. For the former, it depicted a reduced level of initial mitochondria that survived the APAP and/or hypothermic treatment; for the latter, a higher amount of TOM20 signals alone represented de novo synthesis of mitochondria in the sample after APAP and/or hypothermic treatment. In other words, these clearly illustrate a concomitant manifestation of mitophagy and mitochondrial biogenesis, respectively (Figure 3).

### 3.4. Moderate Hypothermic (32 °C) Conditioning Facilitates Effective Hepatic GSH Recycling to Evade a Depletion of ROS Scavenger in AIHI

Aside from mitochondrial study, we further explored the changes in the hepatic GSH content in vitro following hypothermia and APAP toxicity in TAMH. Herein, we demonstrated the capacity of moderate hypothermia to effectively block GSH depletion by advocating a higher rate of GSH recycling from its oxidized GSSG form by approximately two-fold (Figure 4A,B). More importantly, all these occurred notwithstanding a probable retardation in GSH biosynthesis as the transcriptional expression of GCLC, which forms the rate-limiting enzyme in GSH biosynthesis, declined significantly albeit with unchanged GCLM levels (Figure 4C,D).

### 3.5. Moderate Hypothermic (32 °C) Conditioning Suppressed JNK Activation and Subsequent Mitochondrial Translocation to Abate the Amplification of Oxidative Stress in AIHI

For a full-fledged AIHI, the activation of a JNK signaling pathway which amplifies oxidative stress has been frequently reported as an imperative prerequisite [32,33]. Hence, the role of hypothermia in influencing JNK signaling outcomes was inquired. To do so, the level of phosphorylated-JNK(p-JNK)/JNK ratio was determined in distinct subcellular compartments in TAMH. With hypothermic conditioning, a subdued p-JNK/JNK ratio was apparent in both cytosolic and mitochondrial fractions (Figure 5). This may suggest a reduced activation of JNK in the cytosol and consequently, fewer p-JNK translocated across the cytosol into the mitochondria. An observed suppression of the JNK signaling pathway may, therefore, render hypothermia effective in combatting advanced AIHI.

### 3.6. Moderate Hypothermic (32 °C) Conditioning Displayed Cytoprotective Potential in Other DILI with Similar Pathophysiology as APAP Toxicity

Beyond AIHI, we further expanded the scope of investigation to other drug-induced liver injury (DILI) of similar pathophysiology involving mitochondrial dysfunction and oxidative stress [34,35]. Indeed, for both diclofenac- and valproate-induced liver injury, moderate hypothermia could effectively sustain cell viability and avert cell death, comparable with uninjured cells in vitro (Figure 6). The cytoprotection observed in a human hepatocyte model i.e., L-02 cells, further asserts the mechanistic roles of hypothermia in a human system, even though the effects were largely elucidated in mouse hepatocytes i.e., TAMH in this study.

## 4. Discussion

The concept of therapeutic hypothermia for clinical interventions has a longstanding history of more than a century [36]. Amongst which, research efforts to explore its impact in liver injuries have been sporadic, albeit promising hepatoprotection has been observed. Consequently, there is a limited mechanistic understanding of how hypothermia could alleviate liver injuries. Today, facing an outstanding problem of AILI, which ensues the highest incidence of ALF amongst other DILI [3], we made a deliberate attempt to dismiss the uncertainties of hypothermic behavior in AILI. By expounding on the molecular mechanisms underlying hypothermia in AIHI, we aim to clarify and advance understanding, and eventually to foster acceptance of hypothermic therapy for specific applications. In the past work, we precluded a slowdown in metabolism as the major player in hypothermia for attenuating AIHI in a hepatocyte system [22]. While it remains inherent and inevitable in a complex body system, the in vitro AIHI model has allowed the study of other cold-mediated effects, independent of thermodynamics.

Herein, we employed mouse hepatocytes, TAMH, for a detailed in vitro assessment of hypothermic mechanisms in AIHI. This is in continuation from our past work, which demonstrated hepatoprotection with hypothermia in both TAMH and L-02 cell lines [22]. While TAMH is transgenic with human transforming growth factor alpha (TGF-α), to advocate proliferation in the absence of exogenous factors, it retains cytochrome P450 enzyme expressions comparable with wild-type liver or primary mouse hepatocytes and thus displays typical phenotypes of AIHI [37,38,39]. Henceforth, we initiated the investigation in TAMH and highlighted distinct pathways in which hypothermia can act on in APAP toxicity for hepatoprotection—firstly, moderate hypothermia could promote mitophagy to remove damaged mitochondria in AIHI (Figure 1). Corroborating with past reports of autophagic flux following hypothermia [27,40,41], we went further, potentially for the first time, to define a ULK1-independent induction of autophagy in hypothermia. While the ULK1 complex has been well-recognized as an essential component of autophagic initiation [42], recent studies have shown otherwise [43,44]. For example, Cheong et al. has demonstrated a selective manifestation of ULK1/2-independent autophagy in glucose deprivation, but not in amino acid deprivation in vivo [44]. Subsequent reports on glucose starvation has also revealed a crucial role of AMPK to mediate downstream autophagy, dispensable of ULK1, through differential regulation of phosphatidylinositol 3-kinase (Vps34) complexes; one of which is via the phosphorylation of Beclin-1 [45]. Interestingly, AMPK activation was prominent with hypothermia, only in the presence of APAP toxicity even though hypothermia alone could trigger a ULK1-independent autophagy too (Figure 1D). For such contrasting hypothermic behaviors, we lay out some possibilities worthy of future investigations. One, subtle temperature changes, from 37 °C to 32 °C, might be an insufficient stimulus to activate AMPK, a prominent energy sensor which regulates cellular energy homeostasis [46]. With liver injury, in addition to reduced temperature, the bioenergetics may be compromised to a larger extent, sufficient to trigger an AMPK-mediated and ULK1-independent autophagy. Alternatively, AMPK activation in AIHI following hypothermia, may serve additional functions apart from regulating autophagy. Such as in chronic glucose starvation, activated AMPK can stimulate β-oxidation for survival, besides its intuitive association with autophagy [44]. On the contrary, hypothermic conditioning alone, with observed ULK1-independent autophagy, without AMPK or mTOR regulation, might mimic an LC3-associated phagocytosis instead [47]. Further work exploring the translocation of LC3B with lysosomes or phagosomes might be necessary to elicit clearer behaviors of autophagy or phagocytosis respectively. Nonetheless, notwithstanding all these differences, the removal of damaged mitochondria by autophagy with hypothermia in AIHI is apparent (Figure 1A,E). This shapes the role of hypothermia for hepatoprotection when autophagy, and more so mitophagy, has been reported for attenuation of liver injury [26,27].

Secondly, moderate hypothermia could enhance mitochondrial biogenesis in AIHI, concomitant with mitophagy (Figure 2 and Figure 3). While past studies have reported the cold-inducible potential of mitochondrial biogenesis at 4 °C [30,48], we demonstrated its efficacy even at moderate hypothermia of 32 °C in the presence of AIHI. Of note, without APAP toxicity, moderate hypothermia alone may incur an impaired biogenesis despite increased PGC-1α levels (Figure 2). Typically, PGC-1α, as a transcription coactivator, could promote the transcription of nuclear respiratory factor 1 (NRF1) and 2 (NRF2), which in turn increases TFAM expression, and it translocates to the mitochondria to promote mitochondrial DNA replication [49,50]. Yet, the excessive suppression of ΔΨM following hypothermia alone (Figure 2E), could potentially hamper the translocation of newly synthesized TFAM, across the inner mitochondrial membrane by a reduced activation of presequence translocase of the inner membrane (Tim23) [51]. In contrast, an unremarkable change in ΔΨM following hypothermia in AIHI (Figure 2C) could facilitate a smoother execution of mitochondrial biogenesis. Concurrently, it also reflects a comparable mitochondrial activity since ΔΨM is the driving force for ATP synthesis, the central activity in mitochondria [52]. Alongside with reduced protein nitration (Figure 2D) with peroxynitrites [53], hypothermic conditioning effectively curtails mitochondrial dysfunction, a key signature of AIHI. Together, the concomitant interplay of mitophagy and mitochondrial biogenesis could accelerate mitochondria turnover to ensure sufficient levels of functional mitochondria for support of cellular functions. In fact, these simultaneous occurrences of opposing forces may stem primarily from prominent AMPK activation (Figure 1D), especially with known effects of AMPK on promoting mitochondrial mass through PGC-1α [54], besides its eminent autophagic role [55].

Next, moderate hypothermia could allay oxidative stress by blocking GSH depletion and impeding JNK signaling pathway (Figure 4 and Figure 5). In this way, it allows a strict governance of cellular oxidative stress both in the early and late phase of APAP toxicity, respectively. Evidently, the restoration of GSH following hypothermia has been frequently reported, including in clinical studies [56,57,58]. However, little is known regarding the role of hypothermia in GSH homeostasis during AIHI. Therefore, we explored in depth on GSH biosynthesis and recycling process. With a high GSH/GSSG ratio observed during liver injury, GSH recycling appeared to override the potential slowdown in GSH biosynthesis following hypothermia (Figure 4). Amongst the documented effects of hypothermia on GSH recycling, Han et al. demonstrated a greater activity of glutathione reductase (GR) at 4 °C [59] while Slikker et al. described a higher activity of glutathione peroxidase (GPx) in moderate hypothermia [60]. While GR and GPx are opposite forces regulating GSH/GSSG ratio, with the former catalyzing a reduction of GSSG to GSH while the latter scavenge peroxide radicals with GSH to produce GSSG, an increase in activity for either enzymes, following hypothermia, mitigates oxidative stress [61]. In specific AIHI, hypothermic conditioning appears to invoke a greater activity of GR than GPx, if any, to restore the GSH/GSSG ratio and block GSH depletion. In other words, hypothermia enables a sufficient reservoir of GSH to conjugate NAPQI in the early phase of APAP toxicity and furthermore, to scavenge ROS, including superoxides and peroxynitrites, which are responsible for hepatocyte necrosis in AIHI [7,62].

Accompanying GSH restoration, JNK activation was also disrupted with hypothermia in AIHI (Figure 5). With reduced phosphorylation of JNK in the cytosol, there is consequently lesser p-JNK translocated into mitochondria. This hinders the amplification of oxidative stress in mitochondria and prevents the opening of membrane permeability transition (MPT) pore. Ultimately, ATP depletion and subsequent necrotic cell death could, potentially, be evaded [11,63]. For this series of inhibitory effects which hypothermia can exert on the JNK signaling pathway, it was commonly associated with an induction of mitogen-activated protein kinase phosphatase-1 (MPK-1) [64,65]. Being a negative regulator of JNK, MPK-1 has also been shown to protect against AIHI in MPK-1 knock-out mice models [66]. Its apparent hepatoprotective effect following hypothermia may thus warrant further investigations to establish MPK-1 as a cold-inducible protein in AIHI, especially when it can be induced rapidly with other stressors such as UV irradiation and growth factors [67,68]. On the other hand, since JNK signaling may mediate the expression of proinflammatory cytokines [69], the anti-inflammatory potential of hypothermia in the toxic liver can be explored in future studies too. Nevertheless, the association between hypothermia and repressed JNK signaling alone, in the present study, has offered an extra dimension of antioxidant effect, complementary with GSH resolution. Furthermore, a repressed JNK signaling may curtail ROS in the latent phase of APAP toxicity and this may reduce oxidation of GSH to preserve GSH reservoir, besides the involvement of an active GSH recycling process (Figure 4).

Altogether, we have scrutinized the biochemical role of hypothermia in major pathological features encompassing AIHI. This includes the impact of temperature downregulation on mitochondrial dysfunction as well as oxidative stress-mediated pathways (Figure 7). Notably, hypothermia has displayed a consistent characteristic in all aspects investigated—in the absence of liver injury, hypothermia alone presents an unremarkable change; however, with liver injury, there is an adaptive response clearly manifested. This is exemplary of hypothermia as an active process, and not simply a slowdown of general thermodynamics. More so, its capacity for a selective response, depending on the presence of injury, could potentially limit adverse effects following its indiscriminate use as a physical therapy. Finally, with demonstrated effects of hypothermia in mitophagy, mitochondrial biogenesis, GSH recycling and JNK signaling following liver injury (Figure 7), we defined hypothermia as a potential targeted approach for alleviating AIHI. Coupled with its propensity to abate other DILI of similar pathology (Figure 6), encompassing distinct characteristics of reactive metabolite formation alongside mitochondrial dysfunction and/or oxidative stress in diclofenac- and valproate-induced liver injury [35], it further fortifies the mechanisms of hypothermic action unraveled in this study. These promising effects of hypothermia for DILI, therefore, await an extended in vitro study with a larger sample size, *n* ≥ 4, or an in vivo study to cross-validate the observed outcomes. Nonetheless, in the current study, all these may, essentially, serve to rationalize and instill confidence for hypothermia as an unconventional, yet feasible and effective therapy against AILI, and even for other DILIs of similar pathology.

In summary, we hope to transform the hepatoprotective capacity of hypothermia to a clinically actionable therapy for DILI. Hitherto, the therapeutic potential of hypothermia has been most recognized for its neuroprotection in cardiac arrest and neonatal encephalopathy [70,71]. For liver-specific application, while it may be practiced in situ for regional cooling during total hepatic vascular exclusion procedure, the invasive nature of current hepatic cooling techniques may not warrant its use for acute hepatotoxicity [72]. Hence, the outlook hinges on a non-invasive cooling modality for the liver—one which allows timely administration, and potentially localized to evade physiological perturbations associated with systemic cooling. With the elucidation of hypothermic effects in this study, we see a multifaceted behavior of hypothermia, beyond a mere metabolic slowdown. This may redefine hypothermic possibilities beyond conventional continuous systemic cooling employed in neuroprotection. For example, CoolSculpting^®^ (ZELTIQ Aesthetics, Pleasanton, CA, USA) has enabled controlled regional cooling of subcutaneous fats to achieve body shaping effects [73]. This concept of gradual cooling from the skin surface could plausibly be explored for regional non-invasive liver cooling. Until then, hypothermia could be more than an adjunctive therapy, especially for idiosyncratic hepatoxicity with a propensity to worsen in the presence of other pre-existing liver morbidity [74].

## Figures and Tables

**Figure 1 cells-09-02354-f001:**
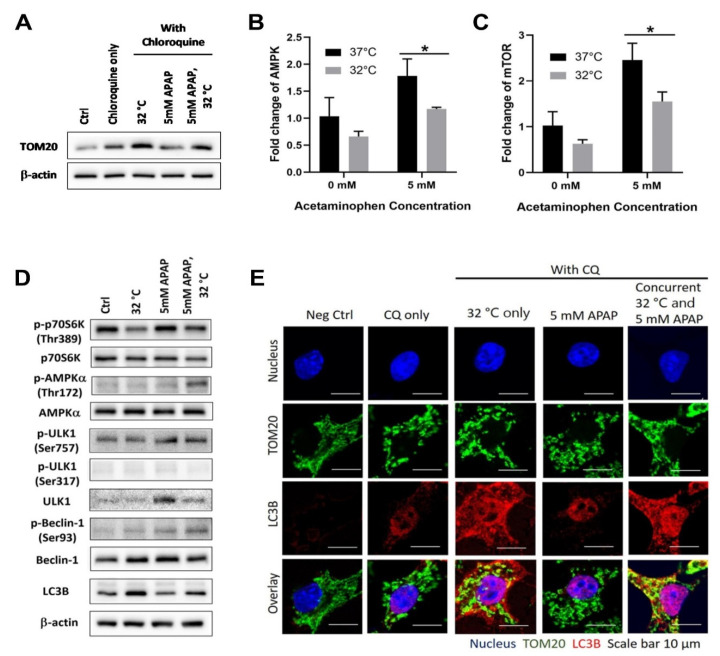
Effect of moderate hypothermia (32 °C) on the extent of mitophagy in transgenic mouse hepatocytes (TAMH). (**A**) Western blotting of microtubule-associated proteins 1A/1B light chain 3B (LC3B) and translocase of outer mitochondrial membrane 20 (TOM20) were performed in the presence of 80 µM chloroquine (CQ), to demonstrate the effect of hypothermia on mitophagy. Western blot for LC3B has been merged to show the relevant bands at the specified sample conditions. Next, quantitative analysis with real time reverse transcriptase-polymerase chain reaction (RT-PCR) was performed to determine the transcript expressions of (**B**) AMP-activated protein kinase (AMPK) and (**C**) mechanistic target of rapamycin (mTOR) while (**D**) western blotting was used to determine the protein expressions of ribosomal protein S6 kinase B1 (p70S6K), phosphorylated-p70S6K (p-p70S6K), AMPKα, phosphorylated-AMPKα (p-AMPKα), unc-51-like autophagy activating kinase 1(ULK1), phosphorylated-ULK1 (p-ULK1), Beclin-1 and phosphorylated-Beclin-1 (p-Beclin-1). (**E**) Immunofluorescence imaging of TOM20 and LC3B was carried out in the presence of 80 µM CQ. β-actin was used as the housekeeping protein for all western blots. Data are presented as mean ± SD (*n* = 3), where an unpaired t-test was used to compare the effect of hypothermia on TAMH against their respective controls treated with the same acetaminophen (APAP) dose. Scale bar = 40 µm. * *p* < 0.05.

**Figure 2 cells-09-02354-f002:**
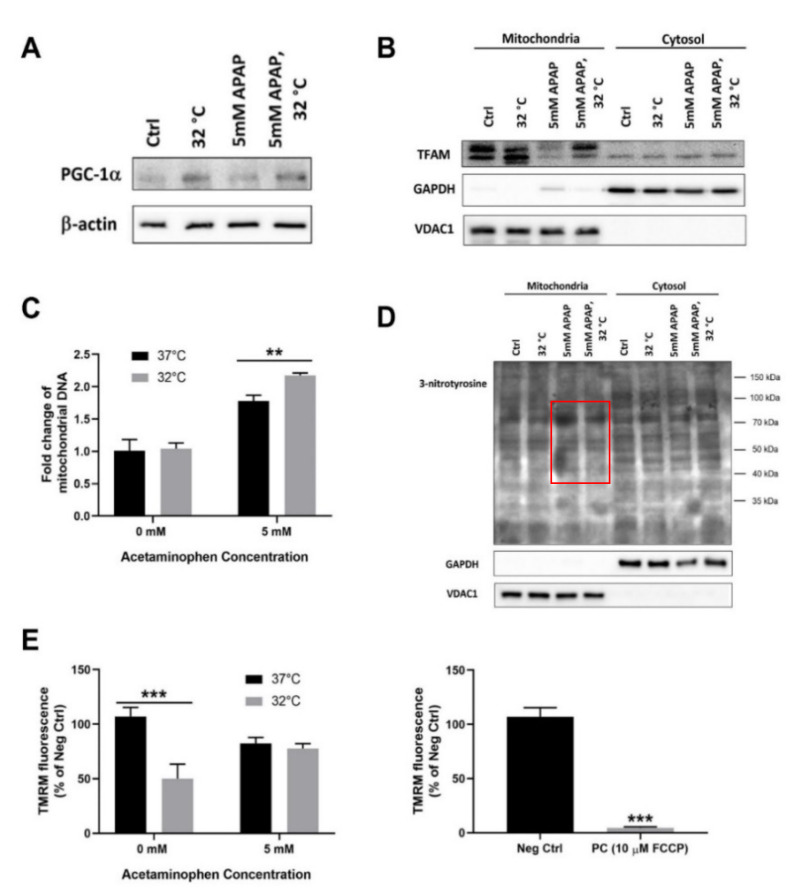
Effect of moderate hypothermia (32 °C) on the extent of mitochondrial biogenesis in TAMH. Western blotting was performed to examine the protein expressions of (**A**) peroxisome proliferator-activated receptor gamma coactivator 1-alpha (PGC-1α) and (**B**) transcription factor A, mitochondrial (TFAM) in the mitochondrial and cytosolic fractions while (**C**) quantitative analysis with real time RT-PCR was carried out to determine the relative mitochondrial DNA levels. (**D**) The extent of mitochondrial dysfunction was also investigated based on the levels of 3-nitrotyrosine protein adducts, where distinct differences were demarcated with the red box and (**E**) the changes in the mitochondrial membrane potential was determined based on the fluorescence intensity of tetramethylrhodamine methyl ester (TMRM); cells treated with 10 µM carbonyl cyanide-p-trifluoromethoxyphenylhydrazone (FCCP) were used as positive control. glyceraldehyde-3-phosphate dehydrogenase (GAPDH) and voltage dependent anion channel 1 (VDAC1) were used as the housekeeping proteins for cytosolic and mitochondrial fractions, respectively, in western blots involving subcellular fractionation while β-actin was used as the housekeeping protein on western blots involving whole cell lysates. Data are presented as mean ± SD (*n* = 3), where unpaired t-test was used to compare the effect of hypothermia on TAMH against their respective controls treated with the same APAP dose. ** *p* < 0.01,*** *p* < 0.001.

**Figure 3 cells-09-02354-f003:**
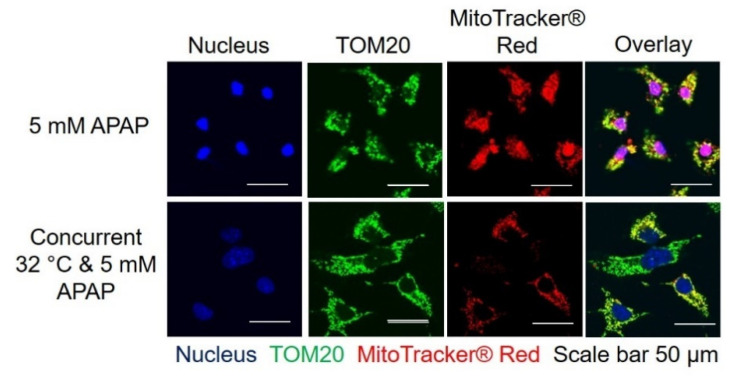
Effect of moderate hypothermia (32 °C) on concomitant mitophagy and mitochondrial biogenesis in the presence of 5 mM APAP-induced hepatocellular injury (AIHI) in TAMH. Immunofluorescence imaging was performed with MitoTracker^®^ Red and TOM20 to track the initial and total number of mitochondria, respectively, prior to and after APAP treatment and hypothermia. Scale bar = 50 µm.

**Figure 4 cells-09-02354-f004:**
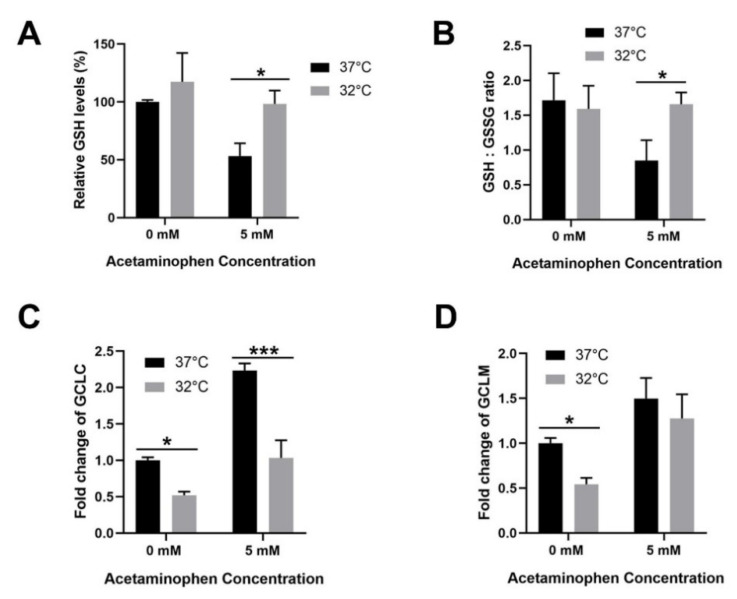
Effect of moderate hypothermia (32 °C) on glutathione homeostasis in TAMH. (**A**) Relative glutathione (GSH) levels and (**B**) GSH/oxidized glutathione (GSSG) ratio were determined to examine GSH changes and the flux of GSH recycling. Quantitative analysis with real time RT-PCR was also performed to determine the transcript expressions of (**C**) glutamate-cysteine ligase, catalytic subunit (GCLC) and (**D**) glutamate-cysteine ligase, modifier subunit (GCLM), which assembles to form the rate-limiting enzyme, glutamate cysteine ligase, in the glutathione biosynthesis. β-actin was used as the housekeeping gene. Data are presented as mean ± SD (*n* = 3), where unpaired *t*-test was used to compare the effect of hypothermia on TAMH against their respective controls treated with the same APAP dose. * *p <* 0.05, *** *p* < 0.001.

**Figure 5 cells-09-02354-f005:**
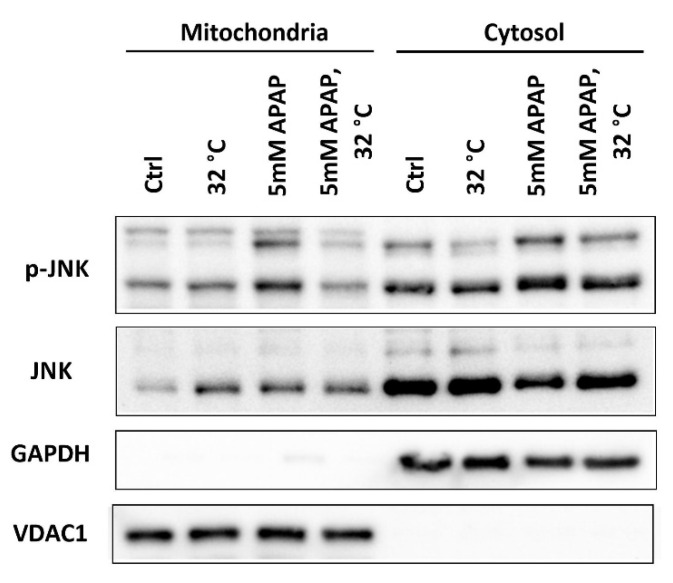
Effect of moderate hypothermia (32 °C) on the activation of the c-Jun N-terminal kinase (JNK) signaling pathway in TAMH. Western blotting was performed to examine the protein expressions of phosphorylated-JNK (p-JNK) and JNK in the mitochondrial and cytosolic fractions. GAPDH and VDAC1 were used as the housekeeping protein for the cytosolic and mitochondrial fractions respectively.

**Figure 6 cells-09-02354-f006:**
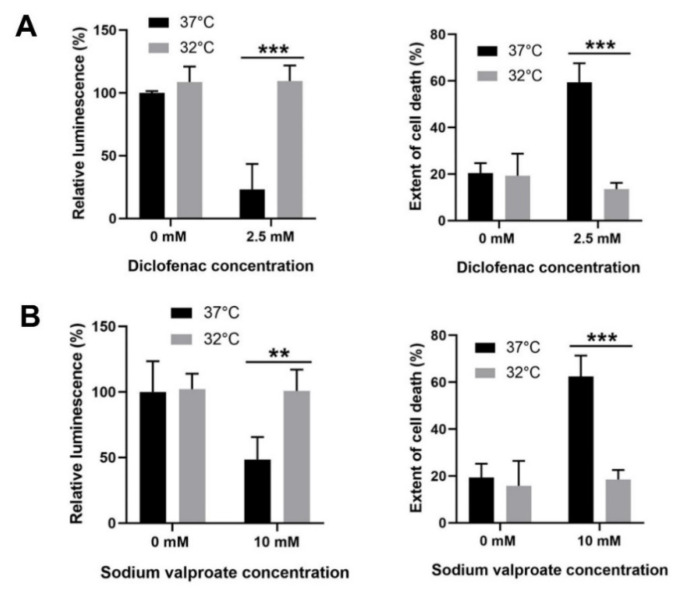
Effect of moderate hypothermia (32 °C) on diclofenac- and valproate-induced liver injury. L-02 cells are subjected to concomitant hypothermic conditioning and (**A**) 2.5 mM diclofenac exposure or (**B**) 10 mM sodium valproate exposure for 24 h. Cell viability was examined with cell titer-glo luminescent assay while cell death analysis was determined with propidium iodide staining. Data are presented as mean ± SD (*n* = 3), where unpaired *t*-test was used to compare the effect of hypothermia on L-02 against their respective controls treated with the same drug dose. ** *p* < 0.01, *** *p* < 0.001.

**Figure 7 cells-09-02354-f007:**
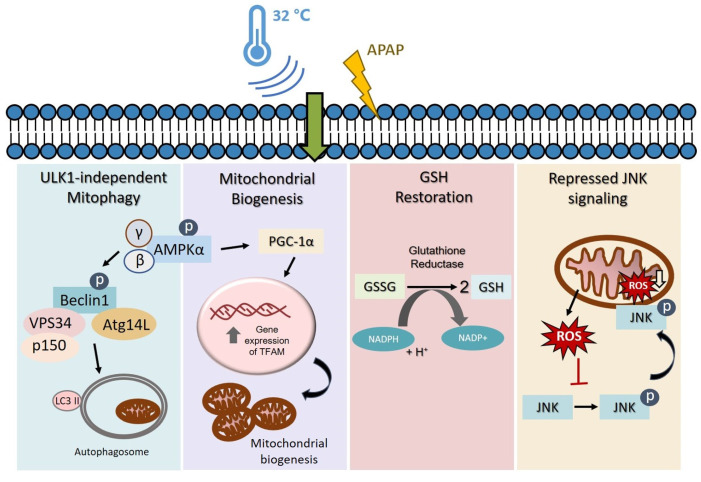
Proposed mechanism of action of moderate hypothermia (32 °C) in attenuating AIHI. Hypothermia conserves the pool of functional mitochondria through concomitant AMPK-mediated mitophagy and mitochondrial biogenesis. The ULK1-independent mitophagy aids in the removal of damaged mitochondria while mitochondrial biogenesis advocates the synthesis of new functional mitochondria during AIHI. Simultaneously, hypothermia could also alleviate oxidative stress by promoting GSH recycling to restore cellular reservoir of GSH, and repressing JNK signaling pathways to impede the subsequent amplification of oxidative stress. All these may play a role to reduce necrotic cell death following AIHI.

**Table 1 cells-09-02354-t001:** Primer sequences used in real-time PCR analysis.

Gene	Accession Number	Primer Sequence (5′ → 3′)
Mechanistic target of rapamycin kinase (mTOR) - homologous to human mTOR	NM_020009.2	F: ATGTGTCCCCCAAACTTCTG R: ATCTTCATGGCCTTTCAGGA
Protein kinase, AMP-activated catalytic subunit alpha 1 (Prkaa1) - homologous to human AMPK	NM_001013367.3	F: AGAGGGCCGCAATAAAAGAT R: TCCTCCGAACACTCGAACTT
Glutamate-cysteine ligase, catalytic subunit (GCLC) -homologous to human GCLC	NM_010295.2	F: TCCATTTTACCGAGGCTACG R: CGATGGTCAGGTCGATGTC
Glutamate-cysteine ligase, modifier subunit (GCLM) - homologous to human GCLM	NM_008129.4	F: CCAGATTTGACTGCCTTTGC R: TGATGATTCCCCTGCTCTTC
Mitochondrially encoded NADH dehydrogenase 1 (ND1) [25] - homologous to human ND1	KY018919.1	F: CTAGCAGAAACAAACCGGGC R: CCGGCTGCGTATTCTACGTT
Hexokinase 2 (HK2) [25] - homologous to human HK2	Y11668.1	F: GCCAGCCTCTCCTGATTTTAGTGT R: GGGAACACAAAAGACCTCTTCTGG
Beta-actin (β-actin)	NM_007393.5	F: TGTTACCAACTGGGACGACA R: GGGGTGTTGAAGGTCTCAAA

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
