# Peer review of "Hypothermia Advocates Functional Mitochondria and Alleviates Oxidative Stress to Combat Acetaminophen-Induced Hepatotoxicity"

_cells, 2020, doi:10.3390/cells9112354_

Round 1
Reviewer 1 Report
The authors of this paper explored the hepatoprotective effects of moderate hypothermia on acetaminophen-induced liver injury, and claimed that it protects via multiple mechanisms including effects on mitophagy, mitochondrial biogenesis, GSH recycling and JNK signaling pathway. While the protective effects of dihydromyricetin were convincingly demonstrated, the proposed mechanisms are questionable. My major concerns are:
-
- It is well established that APAP hepatotoxicity is initiated by P450-mediated metabolic activation. Therefore, examination of the metabolic activation of APAP is the prerequisite before a following protective mechanism can be claimed, since the hepatoprotective effects can entirely result from the inhibited APAP bioactivation by moderate hypothermia. Thus, the effects of moderate hypothermia on APAP metabolism (e.g. by measuring Cyp2e1 expression, early GSH depletion, and NAPQI-protein adducts) should be examined.
- It is known that the cells under a hypothermia condition may exhibit a lower metabolic activity. Is it possible that the cells may just display slower onset of the APAP toxicity under this moderate hypothermia? It would be helpful to show a time course of cell death instead of just relying on a single time point.
- The limitation of this study would be that the whole study relies on a single hepatocyte cell line, while most of those immortalized cell lines cells lack essential drug metabolizing enzymes and transporters that needed for APAP metabolic activation. The caveat is that while those cells still respond to cellular stress such as APAP exposure, its relevance to human pathophysiology is questionable because the nature of the stress is probably different from a cell that can generate APAP reactive metabolite (PMID: 26355817). The study can be greatly improved if similar observations can be validated in the in vivo mouse models, or at least in primary mouse/human hepatocytes.
- In the discussion, it would be desirable if the authors can describe how the hypothermia can be operated and how it can help AILI patients in clinical practice.
Reviewer 2 Report
In this manuscript, the authors found therapeutic hypothermia was protective against multiple modes of drug-induced hepatocellular injury using two in vitro models (TAMH and L-02). Overall, they present a convincing case for the continued investigation of its use in these applications. Furthermore, the authors measure the expression of numerous factors known to contribute to APAP-induced toxicity with positive results. Their results suggest therapeutic hypothermia is worth exploring in the treatment of drug-induced hepatocellular injury. However, I would like to see some revisions prior to publication. One overall comment that I’d like to make: I do have some concerns with the authors using the term AILI in reference to their study results, as they are not studying effects on the liver, rather hepatocellular models. I recognize this is a subtle difference, and while their work certainly has implications as a potential future therapy for AILI, it may be more appropriate if their work was referenced using a term such as acetaminophen-induced toxicity or acetaminophen-induced hepatocellular injury.
Abstract: Well-written. Provides a succinct yet comprehensive overview of their work.
Introduction: I believe it necessary that the authors provide a brief background mitophagy and mitochondrial turnover in this section. Also, the last paragraph of this section should be reworked to improve the flow. The points that the authors are making are sound; however, their thoughts are somewhat scattered and hard to follow.
Results: This section needs a bit of work. While the experimental design is sound and the figures are clear, I really struggled trying to follow the narrative flow. Significant editing needs to be made in this section to improve the readability and limit data interpretation (which should be saved for the discussion).
- Please limit the interpretation of data throughout this section, and simply provide the data. For example: lines 81-86 are an interpretation/contextualization of the results and more appropriately belong in the discussion section.
- Figures should be reference in the order they appear in the text (e.g. Figure 1E is referenced before 1A)
- I may have missed this, but why weren’t the valproate and diclofenac studies also performed in the TAMH line (figure 6)?
Discussion:
- I would like to see more of the author’s thoughts on future directions. How do they expect to transition from their in vitro model to live subjects (animal or human), and what types of limitations do they foresee? I would have also liked to hear more discussion on the current clinical applications of therapeutic hypothermia and the author’s thoughts on the practicality/feasibility of this approach in the clinic.
Materials and Methods: Well-described. Authors use well established techniques.
Round 2
Reviewer 1 Report
I thank the authors for their efforts to address my concerns. However, as the authors stated in the revised manuscript, most of those data I required to add to elucidate my concerns were in the other manuscript that "under review". Without seeing these data, I feel difficult to accept the manuscript in its current form.
